# Infection Dynamics of Swine Influenza Virus in a Danish Pig Herd Reveals Recurrent Infections with Different Variants of the H1N2 Swine Influenza A Virus Subtype

**DOI:** 10.3390/v12091013

**Published:** 2020-09-10

**Authors:** Tarka Raj Bhatta, Pia Ryt-Hansen, Jens Peter Nielsen, Lars Erik Larsen, Inge Larsen, Anthony Chamings, Nicole B. Goecke, Soren Alexandersen

**Affiliations:** 1Geelong Centre for Emerging Infectious Diseases, Geelong, VIC 3220, Australia; anthony.chamings@deakin.edu.au; 2School of Medicine, Deakin University, Geelong, VIC 3220, Australia; 3Department of Veterinary and Animal Sciences, University of Copenhagen, 1870 Frederiksberg C, Denmark; piarh@sund.ku.dk (P.R.-H.); jpni@sund.ku.dk (J.P.N.); lael@sund.ku.dk (L.E.L.); inge@sund.ku.dk (I.L.); nbgo@sund.ku.dk (N.B.G.); 4Division for Diagnostics & Scientific Advice, National Veterinary Institute, Technical University of Denmark, 2800 Kongens Lyngby, Denmark; 5Barwon Health, University Hospital Geelong, Geelong, VIC 3220, Australia

**Keywords:** swine influenza A virus (swIAV), prevalence, H1N2, antigenic diversity, phylogenetic analysis, pig herd

## Abstract

Influenza A virus (IAV) in swine, so-called swine influenza A virus (swIAV), causes respiratory illness in pigs around the globe. In Danish pig herds, a H1N2 subtype named H1N2dk is one of the main circulating swIAV. In this cohort study, the infection dynamic of swIAV was evaluated in a Danish pig herd by sampling and PCR testing of pigs from two weeks of age until slaughter at 22 weeks of age. In addition, next generation sequencing (NGS) was used to identify and characterize the complete genome of swIAV circulating in the herd, and to examine the antigenic variability in the antigenic sites of the virus hemagglutinin (HA) and neuraminidase (NA) proteins. Overall, 76.6% of the pigs became PCR positive for swIAV during the study, with the highest prevalence at four weeks of age. Detailed analysis of the virus sequences obtained showed that the majority of mutations occurred at antigenic sites in the HA and NA proteins of the virus. At least two different H1N2 variants were found to be circulating in the herd; one H1N2 variant was circulating at the sow and nursery sites, while another H1N2 variant was circulating at the finisher site. Furthermore, it was demonstrated that individual pigs had recurrent swIAV infections with the two different H1N2 variants, but re-infection with the same H1N2 variant was also observed. Better understandings of the epidemiology, genetic and antigenic diversity of swIAV may help to design better health interventions for the prevention and control of swIAV infections in the herds.

## 1. Introduction

The influenza A virus (IAV) is a negative-sense, single-stranded, eight-segmented RNA virus belonging to the family *Orthomyxoviridae* [1]. The main antigenic proteins are encoded by the surface gene segments hemagglutinin (HA) and neuraminidase (NA). The six internal gene segments encode for polymerase B2 (PB2), polymerase B1 (PB1), polymerase A (PA), nucleoprotein (NP), matrix (M1 and M2), and non-structural protein (NEP-NS1) respectively [2,3]. There are 18 different HA (H1 to H18) and 11 different NA (N1 to N11) subtypes. Most of these subtypes can be found in aquatic birds (H1-H16 and N1-N9), whereas only a few subtypes are found in mammals [1,4]. In pigs, circulation of IAV, so-called swine influenza A virus (swIAV), is currently mainly limited to three different subtypes including H1N1, H1N2 and H3N2 [5,6,7].

Avian-like swine H1N1 swIAV, was first detected in European pig herds in the late 1970s [8,9] and caused epizootic disease outbreaks that resolved shortly within a few weeks [10,11]. However, several recent studies have shown that the dynamics of swIAV have changed to a more enzootic form where the virus persists in the herds for months or even years [12,13,14,15,16,17]. The altered dynamics of swIAV from a short-term epizootic disease to continuous circulation in the herds, is probably a consequence of increased herd sizes and the continuous supply of naïve individuals that maintain the infection [18,19]. In European pig herds, an average prevalence of 31% has been estimated for swIAV infection [6]. The most prevalent subtypes identified in Europe in recent years were the Eurasian avian-like swine H1N1 (53.6%), the pandemic A/H1N1 2009 (H1N1pdm09) (10.3%), the human-like reassortant swine H1N2 (13%), and the human-like reassortant swine H3N2 (9.1%) [6]. In addition, a number of studies have also shown evidence of reassortants with internal gene segments of H1N1pdm09 and the surface gene segments of predominant enzootic swIAV [20,21,22,23]. In Denmark, the human-like reassortant H1N2 has never been detected. However, another H1N2 reassortant is widespread among Danish pig herds. This subtype is termed “H1N2dk” and has the HA gene of the Eurasian avian-like H1N1 subtype and the NA gene of the H3N2 human-like swIAV [24]. In addition to the subtypes mentioned above, introductions of human seasonal IAV occurs regularly, increasing the risk of novel swIAV reassortants, and making the disease more difficult to control in the herds [23]. In addition, similar to other RNA viruses, swIAV has a high mutation rate, which drives the viral evolution and helps the virus evade the immune system by creating novel variants with modified antigenicity [25,26,27]. Mutations in the antigenic sites (Cb, Sa, Sb, Ca1 and Ca2) of the HA protein can lead to the virus being able to escape the binding of neutralizing antibodies [28,29]. Moreover, mutations in B-cell epitopes and T-cell epitopes of other IAV proteins might also impact the host immunity [30,31,32,33]. Finally, mutations favouring altered *N*-linked glycosylation (NLG) sites near/within HA and NA antigenic sites can also affect the binding of antibodies [34].

According to the Danish Agriculture and Food Council, Denmark produces approximately 30 million pigs annually from around three thousand herds of which approximately 10 million are exported as weaners to other countries such as Poland and Germany [35,36]. In contrast, Denmark imports a very limited number of live pigs. In Denmark and other countries worldwide swIAV is one of the causes of respiratory infection in pigs [37]. SwIAV infections lead to destruction of the epithelial cells and impairs the immune system, thereby making the host more susceptible to infection by other viruses and bacteria. Co-infection of pigs with distinct variants of swIAV and other respiratory pathogens (like *Pasteurella multocida*, *Mycoplasma hyopneumoniae*, *Haemophilus parasuis*, *Actinobacillus pleuropneumoniae*, porcine circovirus type 2 and porcine reproductive and respiratory syndrome virus) are known to cause enhanced disease compared to single pathogen infections, and are all part of the Porcine Respiratory Disease Complex (PRDC), which can lead to massive economic losses [11,38,39,40].

The aim of the present study was to monitor the molecular epidemiology of swIAV circulating in pigs between two weeks of age to 22 weeks of age in a Danish pig herd. A newly developed high-throughput real-time PCR (rtPCR) system (Fluidigm, South San Francisco, USA), which consists of 21 rtPCR assays targeting 18 selected respiratory and enteric viral and bacterial pathogens [41] was applied for initial detection of swIAV in nasal swab samples. Moreover, next generation sequencing (NGS) was used to characterize the complete genome of swIAV during natural infections, and to examine the genetic variability in the antigenic sites of the virus HA and NA proteins.

## 2. Materials and Methods

### 2.1. Ethical Statement

In this study, only nasal swabs were collected and thereby the study did not include the introduction of a needle, which according to the Danish Law on animal experimentation (LBK number 474 of 15 May 2014) is the minimum intervention that requires a specific license. A trained veterinarian was involved in the sampling and data collection, and farmer consent was obtained before the sample collection.

### 2.2. Study Design and Herd Description

The study was designed as an observational cohort study to monitor/screen pig health (particularly swIAV infection) in a Danish farrow-to-finish continuous-flow pig herd. The herd had 985 sows and a farrowing area divided into a number of units. Nursery pigs were housed at a separate site with 2000 pen places, and finisher pigs were raised at a third site with 1050 pen places. According to the Danish SPF (specific pathogen free) herd health declaration (SEGES Svineproduktion 2020), the herd was declared positive for *Mycoplasma hyopneumoniae* (M. hyo), *Actinobacillus pleuropneumoniae* (APP) type 6 and 12 and porcine reproductive and respiratory syndrome (PRRS) Type 2 virus, and negative for APP type 2, PRRS Type 1 virus, *Brachyspira hyodysenteriae*, atrophic rhinitis, Demodex mites and lice.

The herd had contract with a large abattoir and processing company in Denmark, which in 2015 initiated the “Raised Without Antibiotics” (RWA) concept [42]. At present (March 2020), 50 Danish herds including this herd are included in the RWA program, where the producers are payed a premium price to reflect the cost of the increased workload, intensified hygiene measures and other interventions to prevent disease in the effort to reduce the use of antimicrobial agents. The herd produced and recruited gilts internally. After weaning, pigs selected for gilt recruitment were housed in a separate room at the sow site, and vaccinated with a live PRRS Type 2 virus vaccine (Ingelvac^®^ PRRS VET.) at 7 and 10 weeks of age, and additionally with M. hyo, porcine circovirus-2 (PCV2) (Porcilis^®^ PCV M HYO) and swIAV (Respiporc FLU3) vaccines at 7, 10 and 24 weeks of age. All adult sows were vaccinated against swIAV simultaneously four times a year (Respiporc FLU3). Due to clinical signs like nasal discharge and swIAV positive laboratory testing, piglets were vaccinated at four days of age with Respiporc Flu3 (0.4 ml, off label use). During the first week after weaning, pigs were vaccinated against M. hyo and PCV2 (Porcilis^®^ PCV M HYO).

### 2.3. Sample Size Calculation, Sample Collection and Clinical Examinations

The prevalence of different IAV strains in Danish pig herds has been reported to be around 45% [43]. Based on this data, the IAV prevalence in this herd was assumed to be between 10% and 45%. By using a minimum sample size of 40, this should, at a very minimum, give at least one IAV-positive sample with a high probability of getting IAV-positive samples from 4–18 pigs. It was assumed from the outset of the study that some piglets may be lost to follow-up or that some may die and hence it was decided to use an initial sample size of 47. All live-born piglets born within three consecutive farrowing days (n = 518) were ear-tagged at birth with consecutive unique ID numbers. From every 10th piglet (ID number 10, 20, etc.) nasal swabs were obtained (at five to nine day intervals) at week 2 (n = 46), week 3 (n = 38), week 4 (n = 39), week 5 (n = 45), week 6 (n = 44), week 8 (n = 42), week 12 (n = 41) and week 22 (n = 13). Overall, 47 individual piglets/pigs were sampled throughout the study (Appendix A). Due to the variable weaning age (week 4 or week 5) of the piglets, the week 4 sampling was done either at the farrowing unit (week 4 F, n = 24) or in the nursery unit (week 4 N, n = 15) situated at separate buffer stables at the sow site. At week 5, all weaned piglets were moved into separate nursery units and housed with other pigs from the same herd of the same age until approximately 12 weeks of age. Thereafter, the nursery pigs were transferred to a finisher site (~5 kilometre away) where they were housed until slaughter at approximately 22 weeks of age. This finisher site received piglets from the nursery units of the same herd. The samples were collected from the 25 April 2018 until the 20 September 2018. Small- or medium-sized sterile rayon swabs (Medical wire, Wiltshire, UK) were used to collect the nasal swab samples. The swab was inserted into one nostril of the individual piglet and was then turned a full 360°. Samples were stored in 2 mL Phosphate Buffered Saline (PBS) at 4–8 °C until delivered to the laboratory within two days after sampling. At weeks 2 and 4, piglets were clinically examined before obtaining the nasal swabs. Clinical signs of nasal discharge and conjunctivitis were recorded for each individual pig.

### 2.4. Sample Processing and Nucleic Acid Extraction

All samples were processed at the Centre for Diagnostics (CfD), Technical University of Denmark (DTU). The PCR analysis was carried out at CfD-DTU, while the NGS was carried out at Statens Serum Institute (SSI), Denmark. All the collected samples were vortexed for 1 min, the rayon swab was then removed from the tube and 400 µL of the fluid transferred to a 2 mL Eppendorf (EP) tube. The 2 mL EP tubes were centrifuged at 9000× *g* for 5 min at room temperature. Finally, 200µL supernatant was used for nucleic acid extraction. Nucleic acid (both DNA and RNA) was extracted using the QIAcube HT extraction robot (QIAGEN, Hilden, Germany) and the Cador Pathogen 96 QIAcube HT Kit (QIAGEN) according to the manufacturer’s instructions. The extracted nucleic acids were stored at −80 °C for further use.

### 2.5. cDNA Synthesis and Pre-Amplification

The extracted nucleic acids were subjected to reverse transcription using a high capacity cDNA RT Kit (Applied Biosystems, Foster city, CA USA). A final volume of 10 µL reaction mix was prepared by mixing 1 µL of 10X RT buffer, 0.4 µL of 100 mM dNTP mix, 1 µL of 10X random hexamer, 0.5 µL of MultiScribe RT enzyme, 2.1 µL of nuclease free water and 5 µL of extracted nucleic acids. Finally, cDNA synthesis was carried out in a T3 Thermocycler (Biometra, Fredensborg, Denmark) with the given cycling conditions: 25 °C for 10 min, 37 °C for 120 min followed by 85 °C for 5 min and finally paused at 4 °C.

The cDNA sample was pre-amplified using 2X TaqMan PreAmp master mix (Applied Biosystems). A total volume of 10 µL was prepared by mixing 2.5 µL of cDNA with 5 µL of 2X TaqMan PreAmp master mix (Applied Biosystems) and 2.5 µL of a 200 nM primer mix (containing the different sets of primers used for the detection of different pathogens) as previously described [41]. In brief, pre-amplification was carried out in a T3 Thermocycler (Biometra) using the program 95 °C for 10 min followed by 14 cycles of 95 °C for 15 s and 60 °C for 4 min. Finally, the amplification was paused at 4 °C and the pre-amplified product was stored at −20 °C for further use.

### 2.6. High-Throughput Real Time PCR

Initially, the collected samples were screened for IAV by using the high-throughput rtPCR platform BioMark (Fluidigm, South San Francisco, USA) and the 192.24 Dynamic array (DA) integrated fluidic circuit (IFC) chip (Fluidigm). A 4 µL sample mix was prepared by mixing 2.2 µL of pre-sample mix (prepared by mixing 2 µL of 2X TaqMan Gene Expression Mastermix (Applied Biosystem) and 0.2 µL of 20X sample loading reagent (Fluidigm) for each sample) and 1.8 µL of pre-amplified sample. Similarly, 2 µL primer/probe stock was mixed with 2 µL of 2X assay loading reagent (Fluidigm). Three µL of the assay mix and 3 µL of sample mix was loaded into the respective inlets of the 192.24DA IFC chip. The 192.24 DA IFC chip was placed in the IFC controller RX for loading and mixing for approximately 30 min. Finally, the chip was inserted into the high-throughput rtPCR platform BioMark (Fluidigm) for thermal cycling with the following cycling condition: 50 °C for 2 min, 95 °C for 10 min followed by 40 cycles of 95 °C for 15 s and 60 °C for 60 s. All samples were tested in duplicates. Positive and non-template (nuclease-free water) controls were included. Amplification curves and cycle threshold (Ct) values were obtained on the BioMark system and finally analysed using Fluidigm Real Time PCR Analysis software 4.1.3 (Fluidigm) as previously described [41].

### 2.7. RNA Extraction and Real Time RT PCR for the Matrix (M) Gene Assay

Samples found positive for IAV using the high-throughput rtPCR system, were selected for RNA extraction. The extraction was carried out for the selected nasal swabs using the QIAcube extraction robot (QIAGEN) and the RNeasy Mini Kit (QIAGEN) according to the manufacturer’s instructions as described previously [16]. RNA was eluted in 60 µL RNase-free water and stored at −80 °C. Detection of IAV in the extracted RNA was performed in the Rotor-Gene Q (QIAGEN) PCR platform using a previously published rtPCR assay targeting the matrix gene of IAV [15,44].

### 2.8. One-Tube Full Genome IAV PCR and Purification of PCR Product

Confirmed IAV positive samples with Ct values < 30 (using Rotor-Gene PCR) were used for full genome amplification of IAV. A one-tube reaction amplifying each gene segment of IAV was performed using a modified version [45] of a previously published assay [46]. Five µL of the amplified one-tube full genome IAV PCR products along with 10 µL of 1kb DNA ladder (Invitrogen, Carlsbad, CA USA) were run on 0.8% agarose gels (Invitrogen) to check if the bands representing all 8 gene segments of IAV were visible on a Bio-Rad gel documentation system (Hercules, CA, USA). Only fully amplified IAV PCR products were selected and were purified using a high pure PCR product purification kit (Roche, Mannheim, Germany) following the manufacturer’s protocol. The purified PCR products were sent for NGS on the Illumina MiSeq sequencing platform at the State Serum Institute (Copenhagen, Denmark).

### 2.9. Next Generation Sequencing Data Analysis

Sequence analysis was performed in CLC Genomic Workbench 11.0.1. software (QIAGEN). All reads obtained from each of the samples were initially trimmed to remove short and low quality reads and primers/adaptors and then consensus sequences of each gene segment was constructed using the function “Map reads to reference”, using a panel of 22 sequences representing each different lineage of each gene segment known to be present in Denmark. The consensus sequences of each gene segment were then aligned using the MUSCLE algorithm [47], and then examined for similarities using the function “create pairwise comparison”. Moreover, the function, BLASTN, was used to compare the generated consensus sequences with the online NCBI Genbank database [48,49]. Further analysis was done by selecting some of the closest swIAV sequences from the NCBI Genbank. Phylogenetic analysis was done after aligning all sequences of each gene using Clustal-W [50] and using the Maximum Likelihood (ML) method with the best fitting substitution model in MEGA [51]. The HA subtype numbering was done using the Influenza Research Database (IRD) tool at http://www.fludb.org [52,53]. The different amino acids present in the HA antigenic sites (Cb, Sa, Sb, Ca1 and Ca2) were calculated by comparing our sequences with reference sequences [28,54,55,56]. B- and T-cell epitopes were analysed by aligning our sequences with reference sequences [57,58,59,60] using Clustal-W [50]. Differences present at seven antigenic sites (1, 2a, 2b, 2c, 2d, 3 and 4) of the NA gene segment were calculated by comparing our sequences with reference sequences [61,62,63,64].

### 2.10. Analysis of N-Linked Glycosylation Sites of the HA Gene Segment

NetNGlyc 1.0 (http://www.cbs.dtu.dk/services/NetNGlyc/) from DTU Bioinformatics (Department of Bio and Health Informatics) [65] was used for the prediction of *N*-linked glycosylation sites in the HA gene segments only on the N-X-S/T sequons (excluding P at X) with a score threshold >0.5.

### 2.11. Statistical Analysis

Fisher’s exact test was used to determine any association of data using the analysis tool 2by2.xls at http://itve.dk/ [66].

## 3. Results

### 3.1. Prevalence of IAV

The results of the high-throughput rtPCR analysis of nasal swab samples, including an assay for detection of the matrix gene of IAV, allowed for estimation of the prevalence of swIAV at all sampling times. SwIAV was detected in pigs throughout the sampling period starting from week 2 until week 22, indicating continuous circulation of IAV in the herd (Table 1). The prevalence of swIAV increased from the first sampling (week 2) (4.3%) until week 4 (33.3%). Among week-4-old piglets, 41.7% (22.1–63.4% at 95% confidence interval) were IAV-positive in the farrowing unit while 20% (4.3–48.1% at 95% confidence interval) were IAV-positive in the nursery unit (weaned at week 4 or week 5). After weaning (week 5), the prevalence stabilized and then decreased, reaching the lowest prevalence at week 12 (2.4%). After the pigs had been transferred to the finisher site, the prevalence increased again, reaching 30.8% at 22 weeks of age. In total, 76.6% of the pigs were IAV-positive at least once during the study period (Table 1).

### 3.2. Recurrent Detection of IAV in Pigs

Occurrence of IAV is defined as the detection of IAV in nasal swabs in one or more consecutive weeks in the same pig, whereas recurrence is defined as the detection of IAV from the same pig at two or more non-consecutive weeks [67]. Of 47 pigs, 36 (76.6%) were found to be positive for IAV at least once, whereas 11 (23.4%) pigs were found to be negative throughout the study (Appendix A). All the IAV positive and negative pigs were housed together in mixed pens. Among the positives, 7 of the 36 pigs (19.4%) were found to have recurrent IAV (Appendix A), whereas 29 of 36 pigs (80.6%) were found to have only a single occurrence of IAV. Of these 29 pigs, 23 (63.9%) pigs were IAV-positive only at a single sampling time, whereas nine pigs (25%) tested positive at two consecutive sampling time points (Appendix A).

### 3.3. Association Between IAV Infection and Clinical Signs

Among the 46 piglets sampled in week two, 33 (71.7%) had nasal secretion and 20 piglets (43.5%) had conjunctivitis. However, only two (5.1%) piglets had nasal secretion, while four piglets (10.3%) had conjunctivitis in week 4. Nasal secretion was recorded in both of the IAV-positive piglets at week 2, while it was only present in two of the 13 positive piglets at week 4. Similarly, conjunctivitis was observed in one out of two positive piglets at week 2 and was only present in one out of 13 positive piglets at week 4. No association between IAV infection in the individual pigs and clinical signs of nasal secretion and conjunctivitis at week 2 and 4 was observed (Appendix A).

### 3.4. Subtyping of IAV HA and NA Sequences

Fifteen samples that had a Ct value < 30 in the IAV rtPCR assay were selected for one-tube full genome amplification. Of these, 11 samples (Appendix A) displayed clear bands representing all the IAV gene segments on the agarose gel and were selected for IAV whole genome sequencing (WGS) (Appendix A). All the generated sequences for eight full gene segments from all the 11 pig samples have been deposited in NCBI Genbank with accession numbers MT946974 - MT947061.

#### 3.4.1. Analysis of HA Sequences

Eleven full length (1701 nt) HA gene segments from the study were used for phylogenetic analysis. Twenty three additional HA reference sequences representing avian, human, classical swine and H1N1pdm09 subtypes were downloaded from NCBI GenBank and included in the analysis. Phylogenetic analysis revealed that all the 11 HA gene segments were of Eurasian avian-like H1NX origin (1C.2 lineage using the Global Swine H1 Clade Classification System) [68]. All nine H1 sequences obtained from pigs sampled between week 4 and 8 were grouped together in one cluster and had pairwise sequence differences of 0 to 0.6% at the nucleotide level and 0 to 0.8% at amino acid level. This cluster had the highest sequence identity to A/swine/Germany/Holdorf-IDT12357/2010(H1N2) (accession no.: KR699687) in a BLASTN search with ~97% identity at the nucleotide level and ~96.5% at the amino acid level [21]. The two H1 sequences obtained from pigs sampled at week 22 were 100% identical but clustered separately from the H1 sequences obtained from the younger pigs (Figure 1). These H1 genes had the highest sequence identity to A/swine/Denmark/10-1725-1/2011(H1N2) (accession no.: KR700049) in the BLASTN search with identities of 95.4% at the nucleotide level and 96.3% at the amino acid level [21]. As mentioned earlier, sequences from two pigs shedding swIAV at two non-consecutive sampling times were obtained (pig ID 250 sampled at week 4 and week 8, pig ID 380 sampled at week 5 and week 22; Appendix A). In the phylogenetic analysis, the HA sequences of pig ID 250, obtained at weeks 4 and 8 were located in the same cluster and were ~0.5% (9/1701) divergent at the nucleotide level and < 1% (5/566) divergent at the amino acid level. In contrast, the HA sequences of pig ID 380 at weeks 5 and 22 were ~12% (198/1701) divergent at the nucleotide level and ~10% (56/566) divergent at the amino acid level. From the phylogenetic analysis of pig ID 380 it was clearly seen that the HA sequence obtained at the first sampling were located in the cluster defined by the viruses from the younger pigs, whereas the HA sequence obtained at the last sampling (after the pigs had been transferred to the finisher site) were located outside this cluster, thereby representing another H1N2 variant (Figure 1).

The majority of the differences between the two different samplings (week 4 and week 8) for pig ID 250 and week 5 and week 22 for pig ID 380 were located in the antigenic sites of the HA protein between amino acid 78–240. Three different amino acid differences were found in the antigenic sites (V144A and K145N at Ca2 and S160H at the Sa region) of pig ID 250 at week 4 and week 8. Whereas, 16 different amino acids differences were found in the antigenic sites of pig ID 380 at week 5 and week 22 (Table 2). Sequences from different individual piglets sampled at week 4 have one different amino acid (S160H) at the Sa region and two different amino acids (V144A and K145N) at the Ca2 region. Similarly, sequences from pigs of 8 weeks of age had only one different amino acid (S160H) at the Sa region. The two HA sequences from week 22 were found to have identical amino acids in the antigenic sites. The cleavage site with one arginine (PSIQSR: 324–329) and fusion peptide sequence (GLFGAIAGFIEGGWTGMIDGWYG: 330–352) were found to be conserved in all 11 full-length H1 HA sequences [28,69]. The HA2 region of all the HA sequences were found to be more conserved compared to the HA1 region, as they were only approximately 0–9% divergent at the nucleotide level and 0–2% divergent at the amino acid level. However, the HA1 region of all the HA sequences were 0–13.6% divergent at the nucleotide level and 0–14.8% divergent at the amino acid level.

#### 3.4.2. Analysis of NA Sequences

Eleven full-length (1410 nt) NA gene segments were used for phylogenetic analysis (Figure 2). Twenty seven additional NA reference sequences representing Danish H1N2, European H1N2, H3N2 and Asian and American H1N2 NA were also used. The phylogenetic tree shown in Figure 2 revealed that the N2 NA gene segments from the present study were most identical to the NA segment of the Danish H1N2 and European H3N2 subtypes. The N2 sequences obtained from pigs sampled between week 4 and week 8 grouped together in one cluster (Danish HXN2 type) and had pairwise sequence differences of 0 to 0.4% at the nucleotide level and 0 to 0.4% at the amino acid level. This cluster had the highest sequence identity to A/swine/Germany/Holdorf-IDT12357/2010 (H1N2) (accession no.: KR699688) in the BLASTN search with an identity of 97.7% at the nucleotide level and >98% identity at amino acid level, when performing a pairwise comparison [21]. In contrast, the two N2 sequences obtained from pigs at week 22 were identical and were positioned apart from the cluster formed by the sequences obtained from the younger pigs. Similar to the HA segment, this cluster had the highest sequence identity to A/swine/Denmark/10-1725-1/2011 (H1N2) (accession no.: KR700051) in the BLASTN search, with an identity of 96.2% at the nucleotide level and >95% identity at the amino acid level [21] (Figure 2). The N2 sequences obtained from pig ID 250 at weeks 4 and 8 were found to be <0.1% (1/1410) divergent at the nucleotide level and 0.2% (1/469) divergent at the amino acid level. However, the two N2 NA sequences obtained from pig ID 380 at weeks 5 and 22 were found to be ~14.5% (205/1410) divergent at the nucleotide level and ~ 13.6% (64/469) divergent at the amino acid level.

Amino acids present in the antigenic sites (1, 2a, 2b, 2c, 2d, 3 and 4) of the N2 NA sequences of pig ID 250 at week 4 and week 8 were found to be identical. While the N2 sequence obtained from pig ID 380 at week 5 was similar to the ones from pig ID 250, the sequence from pig ID 380 at week 22 were found to be ~35.3% (12/34) divergent in the antigenic sites of N2. Similarly, N2 sequences of pig ID 380 at antigenic site 2d were found to be most divergent, at 75% (3/4), whereas no changes were found in the antigenic site 1 (Table 3). All 11 N2 NA sequences have eight highly conserved amino acids (R118, D151, R152, R224, E276, R292, R371 and Y406) at the inner shell of the NA active site, which interact directly with sialic acids. Similarly, ten highly conserved amino acids (E119, R156, W178, S179, D198, I222, E227, E277, N294 and E425) [61,71,72] were also present in an outer shell of the NA active site. Hence, all the amino acids present at NA active sites were found to be conserved. 

#### 3.4.3. Sequence Comparison of the Internal Gene Segments

All the six internal gene segments (PB2, PB1, PA, NP, M1-M2 and NEP-NS1) were compared with available online NCBI GenBank reference sequences. All the internal gene segment sequences obtained from younger pigs (≤ 8 weeks) were found to be most identical (>98%) with the internal gene segments of H1N1pdm09 origin. In contrast, only four of the internal gene segments (PB2, PB1, PA and NP) of virus sequences from older pigs (week 22) were of H1N1pdm09 origin. The remaining two internal gene segments (M1-M2 and NEP-NS1) of sequences from older pigs (week 22) clustered with the H1N2dk and European H3N2 swIAV subtypes, indicating that these viruses are the result of reassortment events between H1N2-like viruses and the H1N1pdm09 subtypes. In addition, respective internal gene segments obtained from pig ID 250 at week 4 and week 8 were compared and were found to be 0–0.3% different at the nucleotide level and 0–0.4% at the amino acid level. A similar comparison of respective internal gene segments was also done for pig ID 380 at weeks 5 and 22 and were found to be 3.7–22.1% different at the nucleotide level, while at the amino acid level they were 1.2–23.6% different (Table 4). The M1-M2 gene segments of pig ID 380 at week 5 and 22 were found to be 6.5% divergent at the nucleotide level as shown in Table 4. Similarly, the NEP-NS1 gene segments of pig ID 380 at weeks 5 and 22 were found to be most divergent (20.5%) (Table 4) and also clearly supported by two distinct clusters in the phylogenetic tree (Figure 3). NEP-NS1 gene sequences obtained from pigs sampled between week 4 and week 8 made one cluster close to the H1N1pdm09 subtypes. In contrast, the NEP-NS1 sequences obtained from pigs sampled at week 22 made a separate cluster and resembled the H1N2dk and European H3N2 subtypes (Figure 3).

### 3.5. Analysis of B- and T-cell Epitopes Present in swIAV Gene Segments

B- and T-Cell epitopes present in all the swIAV sequences between week 4 and week 8 were identical. One amino acid change (R487K) was observed in the B-cell epitope present in the HA2 part of the HA gene segment after comparing HA sequences from pig ID 380 at week 5 and week 22. Similarly, one (E101D) and four different amino acids (L86F, S91N, E219V and S220T) were found in the T-cell epitope sequence of NP and HA gene segments, respectively. T-cell epitopes of other gene segments were found to be identical among the compared isolates.

### 3.6. Analysis of N-Linked Glycosylation Sites (NLG) of the HA Gene Segments

Six to seven NLG sites were present in all the HA sequences obtained in this study of which five (at positions 21, 33, 276, 483 and 541) were well conserved between all analysed sequences. All the HA sequences obtained from pigs sampled between week 4 and week 8 had one separate NLG site at position 94, whereas the HA sequences obtained from pigs sampled at week 22 have two other NLG sites at position 125–126 and at position 165. The NLG site at position 94 was located near the Ca2 antigenic site in the globular head domain of the HA protein sequence, whereas the NLG site at position 125–126 was located near the Sa antigenic site (Table 5). Similarly, the NLG site at position 165 was located within the Sa antigenic site of the HA protein sequence.

## 4. Discussion

The study was designed to describe the infection dynamics of swIAV in one Danish pig herd by following 47 pigs from 2 weeks of age until slaughter (approximately 22 weeks of age). Using a high-throughput rtPCR system, we were able to determine the prevalence of IAV, and by the use of NGS, we characterized the genetic and antigenic diversity of circulating H1N2 swIAV. Based on the analysis, it was found that the prevalence of IAV in pigs reached a maximum around weaning (4–5 weeks) and then decreased until 12 weeks and then increased again at 22 weeks of age. At least two different H1N2 variants were circulating in the herd, with one of the H1N2 variants circulating at the sow and nursery sites, and the other circulating in the finisher site. Finally, we also demonstrated that individual pigs could have recurrent IAV infections either with a very similar H1N2 variant (pig ID 250) or with two divergent H1N2 variants (pig ID 380). This confirms previous findings that some pigs can have prolonged swIAV infections and be subjected to re-infection, even with closely related swIAV [15,45]. The understanding of the epidemiology and the genetic and antigenic diversity of swIAV in pigs may help to unravel the layers of swIAV infection dynamics and viral evolution from birth to slaughter, thereby helping to design better health interventions for the prevention and control of swIAV in the herds.

The relative low prevalence of IAV before week 4 might be due to the presence of maternally-derived antibodies (MDAs), which in this herd were stimulated by the sows being vaccinated with Respiporc FLU3 four times annually [16,74]. However, the MDAs wane over time, and several studies have shown results indicating that piglets are no longer completely protected from around three to four weeks of age [15,16,74]. Moreover, the loss of MDA occurs at the same time as the piglets are weaned into the nursery, thereby mixing different litters of pigs, creating the optimal environment for IAV circulation. This has also been observed in previous studies [15,16,45]. Similarly, a higher prevalence of IAV at week 4 also indicated that the piglet immune system did not respond to the vaccination at day 4, which is in accordance with a previous study showing no effect of early piglet vaccination against swIAV [45]. One of the explanations for this is that the presence of MDA may hinder an active immune response in the piglets, but it could also be due to the reduced vaccine dose used (0.4ml) [75] in the piglets under study. After week 5, most of the weaned pigs likely developed immunity to the circulating IAV subtype and at the same time no new pigs were introduced into the nursery, resulting in a lower prevalence of IAV in accordance with other findings [76]. The increase in prevalence observed at week 22 at a separate finisher site, was most likely due to infections with a second H1N2dk variant that might have been circulating continuously in the finisher site. This site was managed as a multi-aged, continuous-flow pig herd. This H1N2dk variant from the finisher site differed significantly in antigenic regions from the H1N2dk variant circulating at the sow and nursery sites. Mutations in the antigenic sites of Eurasian avian-like swine H1 have previously been linked to a lack of cross protection and emphasize that the diversity within the subtypes, especially in the H1 avian-like viruses, have now reached a level where it makes no sense to consider viruses of the same subtypes as belonging to the same serotype [15]. However, the lack of a significant cross-reaction should be confirmed by HI-testing, which was not performed in this study. Moreover, the NA gene and the internal gene cassette were also different between the two H1N2dk variants, which could also impact the cross-protective immunity between different swIAV variants of the same subtype [77,78]. Specifically, the internal gene cassette of the H1N2dk variant that infected the pigs in the sow and nursery sites had a complete internal gene cassette of H1N1pdm09 origin, whereas the H1N2dk variant circulating in pigs in the finisher site had an M1-M2 and NEP-NS1 gene of Eurasian avian-like H1Nx origin. However, the protective role of immunity against the internal gene segments are still controversial [58,79].

Pig ID 250 was infected twice with the same H1N2dk variant, which only showed minor genetic differences between samplings. However most of the differences were located in antigenic sites. Hence, it can be speculated that even a small number of mutations could facilitate re-infection with the same subtype, thereby confirming the results of a previous study [45]. In summary, it can be concluded that re-infections can occur with both similar and different variants within the same subtype. The presence of prolonged (2–3 weeks consecutive) and recurrent (non-consecutive) shedding of IAV within week 4 to week 8 pigs also indicated reinfection with the same subtype and this was documented by sequencing. A number of previous studies have shown reinfection with the same strain, leading to prolonged IAV shedding [12,45,74]. The presence of MDA may play role in the prolonged IAV shedding, as MDA may hinder an active immune response [45,80]. Pig ID 380 was infected with two different H1N2dk variants, and most of the mutations were located in antigenic sites as mentioned above. Similarly, the acquisition of NLG sites near/within Sa and Ca2 antigenic sites of the HA sequence may lead to a shielding effect on the antigenic sites and probably the emergence of new antigenic variants. A range of studies have shown that the shielding effect on HA antigenic sites may lead to an evolution of the HA sequence and be responsible for escaping the pre-existing immunity in the hosts [81,82,83,84]. The presence of these major differences between variants within the same subtype emphasizes the presence of a massive genetic drift of Eurasian avian-like H1 in Danish herds [27], which in turn could have consequences for vaccine efficacy as the current swIAV vaccine available against the H1N1 subtype has not been updated since 2002–2003.

In our study, the presence of clinical signs of nasal discharge and conjunctivitis in pigs harbouring IAV at week 4 was not very evident. Reduced levels of clinical signs could be due to the presence of MDA. Similarly, previous exposure or a low level of exposure to the virus might preclude clinical signs in pigs [85,86,87]. In contrast with other studies, our study did not find any association between IAV infection and nasal secretion [15,16]. However, clinical signs were only recorded at two stages (at week 2 and week 4) of the total samplings in this study. Similarly, we did not find any association between IAV infection and conjunctivitis, which is also supported by other studies [15,16].

In conclusion, the complexity of swIAV infection dynamics in pigs from the farrowing unit to the finisher unit has been demonstrated. A high infection pressure of swIAV was identified during the end of the stay in the farrowing unit and the start of the nursery unit. In addition, it has been shown that the prolonged persistence of IAV in pigs could be due to re-infection with IAVs that are closely related to each other. Similarly, re-infections with different strains within the same lineage can also be expected, as the genetic changes affect important antigenic epitopes.

## Figures and Tables

**Figure 1 viruses-12-01013-f001:**
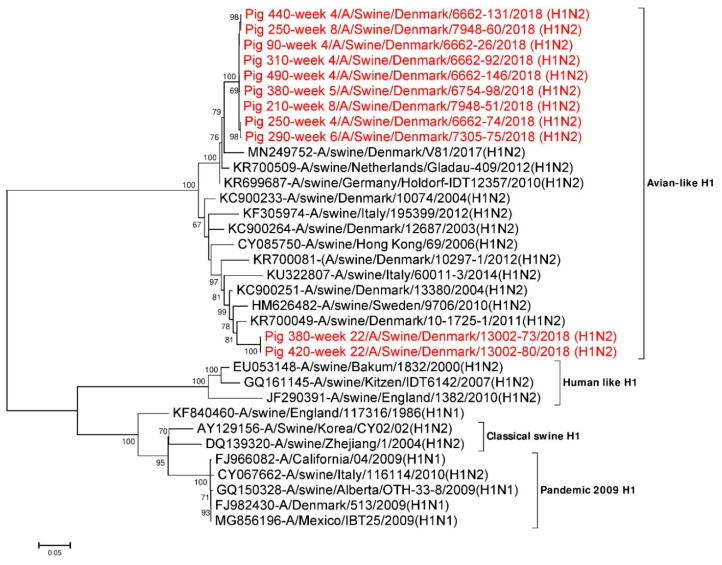
Phylogenetic analysis of the H1 gene segment. The nucleotide sequences were aligned and analysed using the maximum likelihood method in MEGA 7.0 [51], using the General Time Reversible (GTR+G+I) model [70] with a bootstrapping of 1000 replicates. The analysis included 34 H1 sequences of HA gene segments of different IAV H1Nx subtypes, of which 11 H1 gene sequences were obtained in this study (red taxon). The numbers at the nodes represent bootstrap values, and only bootstrap values at or above 60% are shown. Branch lengths are scaled according to the number of nucleotide substitutions per site.

**Figure 2 viruses-12-01013-f002:**
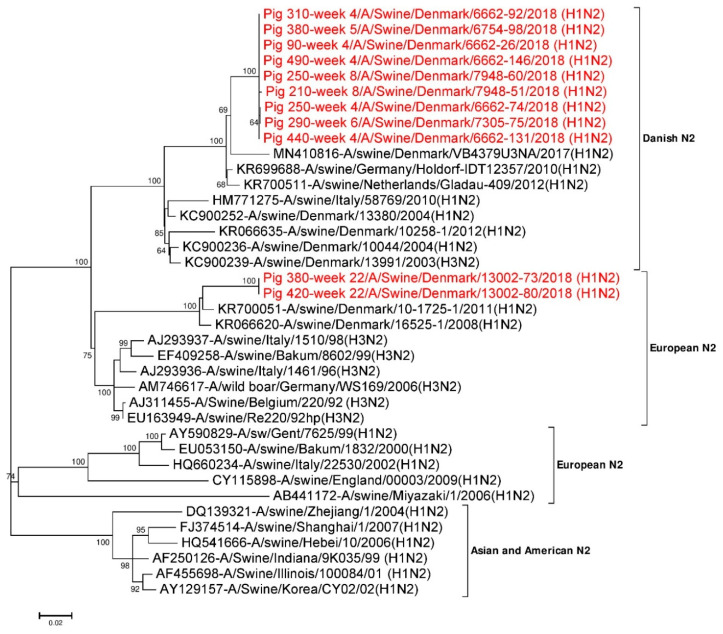
Phylogenetic analysis of the N2 gene segment. The nucleotide sequences were aligned and analysed using the maximum likelihood method in MEGA 7.0 [51], using the General Time Reversible (GTR+G+I) model [70] with a bootstrapping of 1000 replicates. The analysis included 38 N2 sequences of which 11 N2 gene sequences were obtained in this study (red taxon). The numbers at the nodes represent bootstrap values, and only bootstrap values at or above 60% are shown. Branch lengths are scaled according to the number of nucleotide substitutions per site.

**Figure 3 viruses-12-01013-f003:**
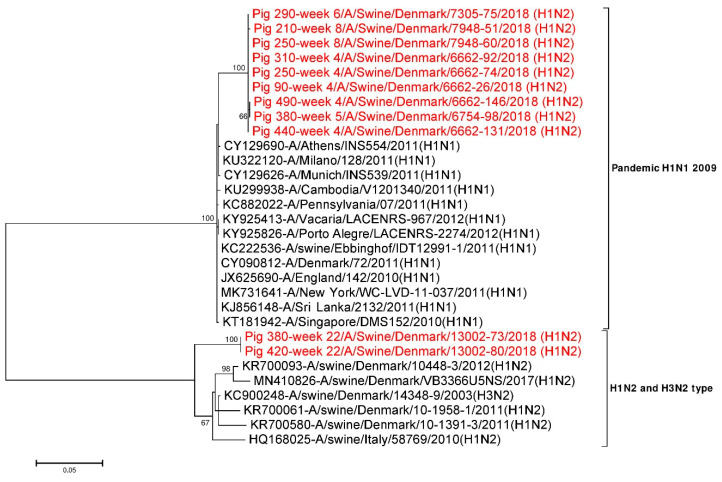
Phylogenetic analysis of the NEP-NS1 gene segment. The nucleotide sequences were aligned and analysed using the maximum likelihood method in MEGA 7.0 [51], using the Tamura three-parameter (T92+G) model [73] with a bootstrapping of 1000 replicates. The analysis included 30 sequences of the NEP-NS1 gene segment of IAV of which 11 NEP-NS1 gene sequences were obtained in this study (red taxon). The numbers at the nodes represent bootstrap values, and only bootstrap values at or above 60% are shown. Branch lengths are scaled according to the number of nucleotide substitutions per site.

**Table 1 viruses-12-01013-t001:** Prevalence of influenza A virus (IAV) in nasal swabs obtained from piglets/pigs at different ages in the herd.

Age of Piglets/Pigs	Prevalence of IAV (with 95% Confidence Interval)
Week 2	4.3% (2/46, 0.5–14.8%)
Week 3	10.5% (4/38, 2.9–24.8%)
Week 4	33.3% (13/39, 19.1–50.2%)
Week 5	31.1% (14/45, 18.2–46.7%)
Week 6	27.3% (12/44, 14.9–42.8%)
Week 8	14.3% (6/42, 5.4–28.5%)
Week 12	2.4% (1/41, 0.06–12.9%)
Week 22	30.8% (4/13, 9.1–61.4%)
Total	76.6% (36/47, 61.9–87.7%)

**Table 2 viruses-12-01013-t002:** Amino acid differences in the hemagglutinin (HA) antigenic sites of pig ID 380 at week 5 and week 22. Amino acid positions were numbered using the H3 numbering system.

Antigenic Sites (H1 HA Gene Segment)	Amino Acid Changes
Cb	S82N
Sa	S156K, R158G, G159N, L164I, S165N, G166N
Sb	Y188D, R189S
Ca1	G173E, S206T
Ca2	F140S, H141Y, A144S, N145K, E225T

**Table 3 viruses-12-01013-t003:** Comparison of amino acid sequences of neuraminidase (NA) antigenic sites of pig ID 380 sampled at week 5 and week 22 from the pig herd. Amino acid positions were numbered using an N2 numbering system.

Antigenic sites (N2 NA Gene Segment)	Amino Acid Change
1	-
2a	N199K
2b	N329D, R331G
2c	K344R, S346D
2d	S367N, N368K, L370S
3	S400R, D402N
4	Q432L, T434D

**Table 4 viruses-12-01013-t004:** Pairwise comparison of six internal gene segments of pig ID 380 sampled at week 5 and week 22 both at the nucleotide and the amino acid level.

Pig ID 380	Gene Segments	Nucleotide Diversity	Amino Acid Diversity
Week 5–Week 22	PB2	5% (114/2280)	2.5% (19/760)
Week 5–Week 22	PB1	4.6% (104 /2274)	1.8% (14/758)
Week 5–Week 22	PA	4.7% (101/2151)	2.5% (18/717)
Week 5–Week 22	NP	3.7% (55/1497)	1.8% (9/499)
Week 5–Week 22	M1-M2	6.5% (64/982)	2.8% (10/351)
Week 5–Week 22	M1	6.9% (52/759)	1.2% (3/253)
Week 5–Week 22	M2	5.4% (16/294)	7.1% (7/98)
Week 5–Week 22	NEP-NS1	20.5% (173/844)	19.8% (68/344)
Week 5–Week 22	NS1	22.1% (146/660)	23.6% (52/220)
Week 5–Week 22	NEP	15.9% (59/372)	12.9% (16/124)

**Table 5 viruses-12-01013-t005:** *N*-linked glycosylation (NLG) sites of H1 hemagglutinin (HA) gene segments obtained from all the pigs from week 4 to week 8 and week 22. “+, ++, +++” indicates the NLG potential with score threshold > 0.5. “*” indicates that the NLG site is located in between amino acid 125–126 for H3 numbering system.

	NLG Sites (Amino Acid Position)										
HA Sequence	NN ST (20)	NS TD (21)	NV TV (33)	NG TC (94)	NE TS* (Between 125–126)	NN SY (165)	NH TY (198)	NC TT (276)	NG TY (483)	NG SL (541)	Total NLG Sites
All pig IDs (week 4 to 8) (Score)	-	+++	+++	++	-	-	-	+	+	++	6
All pig IDs at week 22 (Score)	-	+++	+++	-	++	+	-	+	+	++	7

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
