# Peer review of "Infection Dynamics of Swine Influenza Virus in a Danish Pig Herd Reveals Recurrent Infections with Different Variants of the H1N2 Swine Influenza A Virus Subtype"

_viruses, 2020, doi:10.3390/v12091013_

Round 1

Reviewer 1 Report

The authors should discuss because the flu viruses that infect the pigs when they were younger are different than those that infected then when they were older.

Reviewer 2 Report

The study deals with a virological monitoring of swine influenza A virus (swIAV) infections in a batch of Danish pigs from 2 to 22 weeks of age. The swIAV were then characterised by sequencing. The manuscript is well written and well structured. It is easy to follow. The results are of interest for the scientific community and also for the field to better understand the complexity of swIAV diversity, their evolution and the infections occuring at a pig life level. It is a pity that the immune response of the pigs was not followed to better understand the relationship between the virological and serological status of the pigs and the occurrence of re-infection. Some points of manuscript need clarification before accepting the manuscript for publication. They are detailed in the pdf file in attachment.
